# Photoinhibition and Photoprotective Responses of a Brown Marine Macroalga Acclimated to Different Light and Nutrient Regimes

**DOI:** 10.3390/antiox12020357

**Published:** 2023-02-02

**Authors:** Hikaru Endo, Hikari Moriyama, Yutaka Okumura

**Affiliations:** 1Faculty of Fisheries, Kagoshima University, Kagoshima 890-0056, Japan; 2United Graduate School of Agricultural Sciences, Kagoshima University, Kagoshima 890-0056, Japan; 3Fisheries Resources Institute/Fisheries Technology Institute, National Research and Development Agency, Japan Fisheries Research and Education Agency, Shiogama 985-0001, Japan

**Keywords:** antioxidant, carotenoid, chlorophyll fluorescence

## Abstract

Plants and brown algae avoid photoinhibition (decline in photosystem II efficiency, *Fv*/*Fm*) caused by excess light energy and oxidative stress through several photoprotective mechanisms, such as antioxidant xanthophyll production and heat dissipation. The heat dissipation can be measured as non-photochemical quenching (NPQ) and is strongly driven by de-epoxidation of xanthophyll cycle pigments (XCP). Although NPQ is known to increase under high light acclimation and nutrient-deficient conditions, a few studies have investigated the combined effects of the conditions on both NPQ and associated xanthophyll-to-chlorophyll (Chl) a ratio. The present study investigated the photosynthetic parameters of the brown alga *Sargassum fusiforme* acclimated to three irradiance levels combined with three nutrient levels. Elevated irradiance decreased *Fv*/*Fm* but increased NPQ, XCP/Chl *a* ratio, and fucoxanthin/Chl *a* ratio, suggesting the photoprotective role of antioxidant fucoxanthin in brown algae. Reduced nutrient availability increased NPQ but had no effect on the other variables, including XCP/Chl *a* ratio and its de-epoxidation state. The results indicate that NPQ can be used as a sensitive stress marker for nutrient deficiency, but cannot be used to estimate XCP pool size and state.

## 1. Introduction

In photosynthetic organisms, high irradiance (i.e., high light intensity) under environmental stresses, such as low and high temperature, often causes photoinhibition, which is described as a decline in photosystem II efficiency (*Fv*/*Fm*), because excess light energy enhances the production of harmful reactive oxygen species in chloroplasts [1,2,3]. To cope with the oxidative stress, such organisms have evolved several photoprotective mechanisms, including the production of antioxidants and dissipation of the excess light energy as heat [2,3,4]. Both mechanisms are strongly associated with carotenoids (carotenes and xanthophylls) because several compounds, such as β-carotene (β-Car), zeaxanthin (Zx), and fucoxanthin (Fx) act as antioxidants [3,5,6], and the de-epoxidation of xanthophyll cycle pigments (XCP) from violaxanthin (Vx) to Zx through antheraxanthin (Ax) is one of major drivers of the heat dissipation in various organisms, including plants and brown algae [4,7,8].

As one of the pigments involved in the carotenoid biosynthetic pathway that leads to the formation of xanthophylls from carotene, β-Car can be converted to β-cryptoxanthin, Zx, Ax, Vx, and neoxanthin, while Fx in brown algae is hypothesized to be synthesized from neoxanthin [6]. Some of these carotenoids bind with protein and chlorophyll (Chl) *a*, constituting photosystem II reaction center [9] and light-harvesting complex II [10]. The carotenoid-to-Chl *a* ratio of various organisms is reported to increase in response to high light acclimation [11,12,13]. Moreover, our previous studies have shown the effects of elevated irradiance and nutrient availability on Chl *a* content, Chl *c*/ Chl *a*, Fx/Chl *a*, and XCP/Chl *a* ratios in brown algae [14,15]. However, the role of Fx as a photoprotective compound in brown algae remains unclear because the Fx/Chl *a* ratio does not change under elevated irradiance acclimation [14,15].

The heat dissipation can be quantified as non-photochemical quenching (NPQ) using pulse amplitude modulation (PAM) chlorophyll fluorometers [4,5,6]. Previous studies have shown that high light acclimation enhanced NPQ (or the similar indicator, qN) of several organisms, including brown algae [4,16,17,18]. Moreover, NPQ is reported to increase under nutrient-deficient conditions in plants [19,20] and algae [17,21,22,23], although the underlying physiological mechanisms remain uncertain [14]. Meanwhile, NPQ generally correlates with de-epoxidation states (DES) of XCP in plants [24] and brown algae [11], and it is also strongly affected by the total amount of XCP (i.e., XCP pool size) in brown algae [4,11]. Hence, increased NPQ of brown algae under reduced nutrient availability may be caused by changing XCP pool size and/or DES of XCP. However, the nutrient effects on NPQ and XCP of brown algae have rarely been quantified at the same time.

Large brown algae (kelps and fucoids) are the dominant taxa in temperate reef ecosystems [25,26,27]. The present study investigated the combined effects of irradiance and nutrient availability on photosynthetic parameters (*Fv*/*Fm* and NPQ) and pigments (Chl *a* content, Chl *c*/ Chl *a* ratio, β-Car/Chl *a* ratio, Fx/Chl *a* ratio, XCP/Chl *a* ratio, and DES) in the fucoid brown alga *Sargassum fusiforme*, which is one of the common and commercially important species in Asian countries. Holdfasts of the species were used in the present study instead of the shoots because they have important ecological traits, including the ability to grow without shoots, regeneration ability via germination of new shoots (i.e., vegetative reproduction) [28,29], and high tolerance to warm and nutrient-poor conditions [28].

## 2. Materials and Methods

### 2.1. Sample Preparation

Five individuals of *S. fusiforme* with relatively large holdfasts were collected in June 2021 at a depth of 0–1 m along the Yojiro coast (31°33′30″ N, 130°33′47″ E), Kagoshima Bay, southern Japan, and were transported to the laboratory in insulated cool boxes. Ten holdfast segments (5 mm in length) without shoots and other sessile organisms were cut from each specimen, yielding a total of 50 segments. Each of the 10 segments derived from the *S. fusiforme* individuals were placed in petri dishes containing 30 mL sterilized seawater (a total of five dishes). Subsequently, the petri dishes were incubated for 24 h at an optimal temperature of 20 °C [28] and irradiance of 130 μmol photons m^−2^ s^−1^ with a 12 h light (L): 12 h dark (D) photoperiod.

### 2.2. Culture Experiment

At the beginning of the experiment, the wet weight of each segment (initial value) was measured using an electronic balance (0.1 mg accuracy) after the removal of excess moisture by blotting on paper towels. The 50 segments were divided into 10 groups, with each group having five specimens with a similar size distribution. One of the 10 groups was used to evaluate initial photosynthetic parameters and pigment content. Each of the remaining nine groups were subjected to one of the nine different treatments, consisting of three irradiance levels (30, 130, and 300 μmol photons m^−2^ s^−1^) and three nutrient levels.

The three irradiance levels were set based on compensation (i.e., 5–37 μmol photons m^−2^ s^−1^) [30], optimal growth (i.e., 100–180 μmol photons m^−2^ s^−1^) [31], and saturation irradiance values of the species (i.e., 391 μmol photons m^−2^ s^−1^) [30]. Light with wavelength of 400–700 nm (i.e., photosynthetically active radiation) was provided by white fluorescent tubes. The nutrient levels were achieved by adjusting the dilution ratio of Provasoli’s enriched seawater (PESI) to 0%, 5%, and 25%. The 0% PESI was prepared using artificial seawater without nitrates or phosphates (Marine art SF-1, Tomita Pharmaceutical Co., Ltd., Naruto, Japan). The dissolved inorganic nitrogen concentrations in the 5% and 25% PESI are theoretically 40 and 200 μM, respectively [14,15]. 

The holdfast segments were cultured for 28 d at 20 °C under a 12 h L: 12 h D photoperiod. The culture media in each petri dish were changed every 7 d. Each treatment was replicated using five segments derived from five different individuals (one segment per petri dish, five petri dishes per treatment) because physiological responses may differ among individuals [32]. At the end of the culture experiment, the wet weights of the segments (final value) were measured and relative growth rates (RGR, % d^−1^) were calculated as 100 × ln(final value/initial value)/28 d. The *Fv*/*Fm* of each segment was measured using a JUNIOR-PAM chlorophyll fluorometer (Heinz Walz GmbH, Effeltrich, Germany) after more than 1 h of dark acclimation to exclude energy-dependent fluorescence quenching and state transitions [33,34]. The maximum value of NPQ was also measured during the exposure of segments to actinic light intensity of 1150 μmol photons m^−2^ s^−1^ for 5 min.

### 2.3. Pigment Analyses

Pigment contents, including Chl *a*, Chl *c*_2_, β-Car, Zx, Ax, Vx, and Fx were measured using high-performance liquid chromatography (HPLC) after transferring the samples into 6 mL bottles containing 4 mL dimethylformamide. The samples were diluted in distilled water to achieve 80% concentration and then analyzed by an HPLC system (Shimadzu Corp., Kyoto, Japan), according to the method described by Zapata et al. [35]. Afterward, the total content of XCP (the sum of Vx, Ax, and Zx) and DES (the ratio of 0.5Ax + Zx to total XCP) [10] were calculated. Only Chl *a* content and the ratios of other pigments (Chl *c*_2_, Fx, and XCP) to Chl *a* in *S. fusiforme* holdfasts are presented in this study because the responses to abiotic factors among the pigment contents were similar; however, the responses differed among the ratios of the pigments to Chl *a* content of the kelp *Undaria pinnatifida* [14].

### 2.4. Statistical Analyses

Differences in the initial wet weights of *S. fusiforme* segments among the ten treatment groups were analyzed using analysis of variance (ANOVA). The effects of irradiance and nutrient levels on all variables of the holdfasts were analyzed by two-way ANOVA. Some variables were logarithmically transformed because the data were not normally distributed (Shapiro-Wilk test, *p* < 0.05) and did not show homogeneous variances (Bartlett’s test, *p* < 0.05). All analyses were performed using R (https://www.r-project.org, accessed on 1 November 2022). Data are presented as means ± standard deviation.

## 3. Results

The average initial wet weight of *S. fusiforme* holdfasts was 12.0 ± 4.07 mg. No significant differences were observed in the initial wet weights among the ten groups (df = 9, MS =13.670, F = 0.790, *p* = 0.627). The initial *Fv*/*Fm* and NPQ values were 0.67 ± 0.06 and 3.25 ± 0.60, respectively. The initial values of Chl *a* content, Chl *c*/Chl *a* ratio, β-Car/Chl *a* ratio, Zx/Chl *a* ratio, Vx/Chl *a* ratio, Fx/Chl *a* ratio, XCP/Chl *a* ratio, and DES were 0.201 ± 0.029 mg/g, 0.081 ± 0.007, 0.068 ± 0.005, 0.002 ± 0.001, 0.051 ± 0.005, 0.246 ± 0.014, 0.053 ± 0.005, and 0.039 ± 0.016, respectively. 

According to the two-way ANOVA results, irradiance had a significant effect on RGR, *Fv*/*Fm*, Chl *a* content; however, the effect on Chl *c*/Chl *a* ratio was not significant (Table 1 and Figure 1). RGR was significantly higher at medium and high irradiance levels than at low irradiance levels. In contrast, *Fv*/*Fm* was lower at medium and high irradiance levels than at low irradiance levels. Chl *a* content decreased in response to elevated irradiance. The effects of nutrient levels and interaction between irradiance and nutrient levels on the variables were not detected.

Irradiance had no effect on the β-Car/Chl *a* ratio, but had an effect on the Zx/Chl *a*, Vx/Chl *a*, and Fx/Chl *a* ratios (Table 1 and Figure 2). The Zx/Chl *a* ratio was significantly higher at medium and high irradiance levels than at low irradiance levels. The Vx/Chl *a* and Fx/Chl *a* ratios increased in response to elevated irradiance. The effects of nutrient levels and interaction between irradiance and nutrient levels on the variables included in the analysis were not detected. Additionally, high correlations were observed between the β-Car/Chl *a* and Zx/Chl *a* ratios, and between the Vx/Chl *a* and Fx/Chl *a* ratios, but not between the Zx/Chl *a* and Vx/Chl *a* ratios, and between the Zx/Chl *a* and Fx/Chl *a* ratios (Figure 3).

Irradiance had a significant effect on the XCP/Chl *a* ratio and NPQ (Table 1 and Figure 4); however, its effect on DES was not significant (Table 1). The XCP/Chl *a* ratio increased in response to elevated irradiance. NPQ was significantly higher at medium and high irradiance levels than at low irradiance levels. Nutrient levels had an effect on NPQ only, and NPQ was higher in 0% PESI treatments than in the 5% and 25% PESI treatments. No significant interaction was observed between irradiance and nutrient levels on the variables included in the analysis.

## 4. Discussion

The indicator of photoinhibition, *Fv*/*Fm* of photosynthetic organisms is known to decrease in response to high irradiance under environmental stresses, such as high or low temperature [1,2,3]. For example, Balfagón et al. [36] showed that a negative effect of increased irradiance (50–600 μmol photons m^−2^ s^−1^) on *Fv*/*Fm* of the terrestrial plant *Arabidopsis thaliana* was synergized by elevated temperature from 23 °C to 42 °C. Similarly, Xu et al. [29] reported that *Fv*/*Fm* of the brown algae *S. fusiforme* reduced in response to increased irradiance (70–140 μmol photons m^−2^ s^−1^) at a high temperature of 24 °C but not at 16–20 °C. Machalek et al. [37] also reported that *Fv*/*Fm* of the kelp *Saccharina latissima* decreased in response to elevated irradiance (15–150 μmol photons m^−2^ s^−1^) at a low temperature of 5 °C but not at 17 °C. Meanwhile, nutrient-poor conditions during summer are one of the environmental stresses for brown algae, including *S. fusiforme* holdfasts [28,38]. In the present study, *Fv*/*Fm* of *S. fusiforme* holdfasts decreased by elevated irradiance (30–300 μmol photons m^−2^ s^−1^), while no significant interaction between irradiance and nutrient conditions was observed. This result suggests that the negative effect of high irradiance on *Fv*/*Fm* of this species was not strengthened by nutrient-poor conditions.

Balfagón et al. [36] also showed that the increased irradiance combined with the elevated temperature caused not only decreased *Fv*/*Fm* but also leaf senescence and reduced survival rate of *A. thaliana*. Similarly, Endo et al. [39] reported that elevated temperature (23–26 °C) at high irradiance (180 μmol photons m^−2^ s^−1^) resulted in reduced *Fv*/*Fm* and blade erosion of the kelp *Eisenia bicyclis*. Thus, photoinhibition is often accompanied by physiological damages in plant and algal bodies. However, the elevated irradiance (30–300 μmol photons m^−2^ s^−1^) reduced *Fv*/*Fm* but enhanced growth of *S. fusiforme* holdfasts in the present study. Therefore, the decline of *Fv*/*Fm* under high light acclimation seem not necessarily to accompany growth suppression in the absence of environmental stresses.

Chl *a* content of plants and algae is known to decrease under high light acclimation [14,15,40,41] and this response contributes to limiting absorption of excess light energy [19]. In addition, in the present study, Chl *a* content of *S. fusiforme* holdfasts decreased in response to increased acclimation irradiance from 30 to 300 μmol photons m^−2^ s^−1^. Moreover, microalgae are known to decrease their ratio of accessory Chl to Chl *a* under high light acclimation [41]. Previous studies have shown that Chl *c*/Chl *a* ratio of brown algae was lower in the thalli grown at a sun-exposed shallower depth than the same species collected at greater depths or shaded site [42]. However, in the present study, no significant difference was found in Chl *c*_2_/Chl *a* ratio of *S. fusiforme* holdfasts acclimated in three different irradiance conditions. Charan et al. [15] also reported that the Chl *c*_2_/Chl *a* ratio of *S. fusiforme* shoots was not affected by irradiance at 23 °C, although the value decreased in response to increased irradiance (30–150 μmol photons m^−2^ s^−1^) combined with heat stress (26 °C). Moreover, Endo et al. [14] reported that increased irradiance (30–180 μmol photons m^−2^ s^−1^) caused a decrease in the Chl *c*_1_/Chl *a* ratio but an increase in the Chl *c*_2_/Chl *a* ratio of the kelp *U. pinnatifida*. Hence, the variation in the Chl *c*/Chl *a* ratio of brown algae along depth and sun-exposure gradients found in previous studies [42] cannot be explained by the response to changing irradiance. It might be affected by changes in light quality associated with depth or sun-exposure, although this hypothesis needs to be tested.

In the carotenoid biosynthetic pathway, β-Car can be converted to β-cryptoxanthin, Zx, Ax, Vx, and neoxanthin, while Fx in brown algae is hypothesized to be synthesized from neoxanthin [6]. The hypothesis is partially supported by the results of the present study, which showed a stronger correlation between the Vx/Chl *a* and Fx/Chl *a* ratio than between the Zx/Chl *a* and Vx/Chl *a* or Fx/Chl *a* ratios in *S. fusiforme* holdfasts acclimated to nine types of treatments. In addition, a strong correlation was observed between the β-Car/Chl *a* and Zx/Chl *a* ratios, nevertheless elevated irradiance did not affect β-Car/Chl *a* ratio but affected Zx/Chl *a* ratio, implying that β-Car might be rapidly converted to Zx in this alga. Xie et al. [12] reported that elevated irradiance increased β-Car content of the red alga *Neopyropia yezoensis*, while such an increase in β-Car/Chl *a* ratio was not observed in the brown alga *S. fusiforme* in the present study. In contrast, irradiance elevation did not affect the Fx/Chl *a* ratio of *U. pinnatifida* and *S. fusiforme* shoots but increased the ratio of *S. fusiforme* holdfasts in the present study, suggesting that Fx is one of photoprotective compounds in this alga. Hence, the chemical form of carotenoids accumulated as a result of high light acclimation may differ among species and parts within the species.

Ocampo-Alvarez et al. [11] observed that the blades of the kelp *Macrocystis pyrifera* acclimated to shallow depths exhibited higher XCP/Chl *a* ratios and NPQ than those grown at greater depths. Moreover, previous studies have reported that high light acclimation resulted in increases in XCP/Chl a ratios [14] and NPQ [18] of brown algae. In addition, in the present study, the elevated irradiance increased both XCP/Chl a ratios and NPQ of *S. fusiforme* holdfasts. Hence, increased NPQ under high light acclimation can be explained by increased XCP pool size of the brown algae. Meanwhile, NPQ decreased in response to nutrient enrichment in *S. fusiforme*, which is consistent with the observation made in plants [19,20] and algae [17,21,22,23]. Although the XCP/Chl a ratio can decrease under nutrient enriched conditions [14,15], XCP/Chl *a* ratios and DES observed in our study did not change. The results suggest that the decrease in NPQ in response to nutrient enrichment was not due to the decreases in XCP pool size and de-epoxidation rate of Vx to Zx. Therefore, NPQ can be used as a sensitive stress marker for nutrient deficiency but cannot be used to estimate XCP pool size and state. Such a reduction in NPQ without any changes in XCP might be associated with state transitions between photosystems I and II [43,44], although this possibility needs to be examined.

## 5. Conclusions

The results of the present study showed that elevated irradiance caused photoinhibition and Chl *a* degradation but enhanced growth and photoprotection, which were reflected by the increases in Fx/Chl *a* ratios, XCP/Chl *a* ratios, and NPQ in *S. fusiforme* holdfasts, while nutrient availability had negligible effects on the variables other than NPQ. These traits may enhance the survival and growth of *S. fusiforme* holdfasts, without forming a canopy of its own shoots under high irradiance and nutrient-poor conditions that occur during summer [28]. However, little is known regarding the combined effects of elevated summer temperatures under climate change and increased irradiance on XCP/Chl *a* ratios and NPQ in brown algae, including *S. fusiforme*. Furthermore, the mechanisms of associated with decreases in NPQ under nutrient enrichment conditions still remain unknown. Therefore, further studies should be conducted to investigate the effects of abiotic factors on NPQ and associated variables to enhance our understanding of photoprotection in photosynthetic organisms, including brown macroalgae.

## Figures and Tables

**Figure 1 antioxidants-12-00357-f001:**
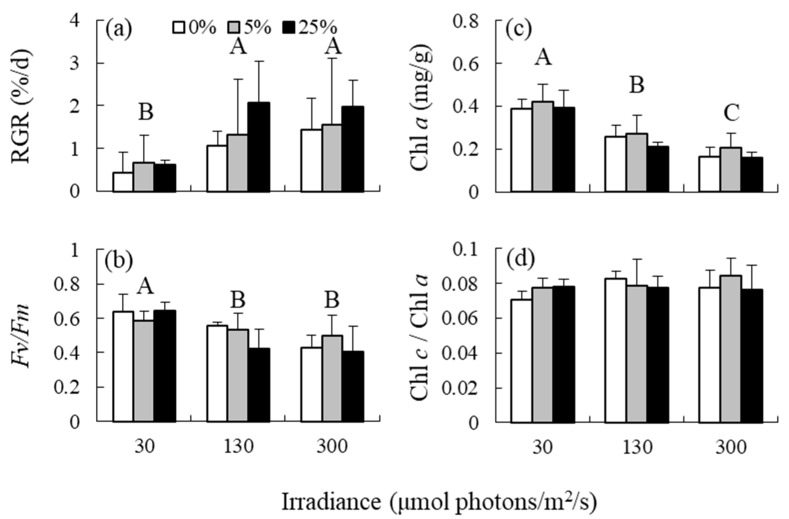
(**a**) Relative growth rates (RGRs), (**b**) photosystem II efficiency (*Fv*/*Fm*), (**c**) chlorophyll (Chl) *a* content, and (**d**) Chl *c*_2_/Chl *a* ratios of the brown alga *Sargassum fusiforme* acclimated to three irradiance levels (30, 130, and 300 μmol photons m^−2^ s^−1^) and three nutrient levels (0, 5, and 25% Provasoli’s enriched seawater, PESI). Values are expressed as means ± standard deviation. Different large letters indicate statistical significances (*p* < 0.05) among the different treatments.

**Figure 2 antioxidants-12-00357-f002:**
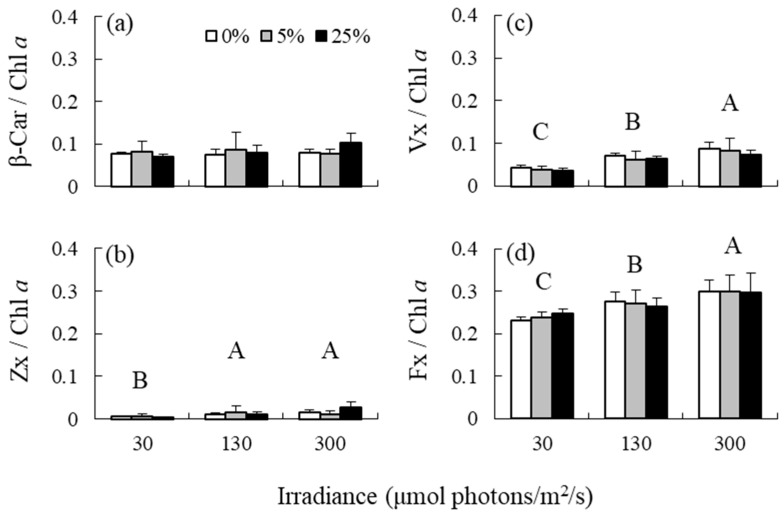
(**a**) β-carotene (β-Car)/Chl *a* ratios, (**b**) zeaxanthin (Zx)/Chl *a* ratios, (**c**) violaxanthin (Vx)/Chl *a* ratios, and (**d**) fucoxanthin (Fx) /Chl *a* ratios of the brown alga *Sargassum fusiforme* acclimated to three irradiance levels (30, 130, and 300 μmol photons m^−2^ s^−1^) and three nutrient levels (0, 5, and 25% Provasoli’s enriched seawater, PESI). Values are expressed as means ± standard deviation. Different large letters indicate statistically significant differences (*p* < 0.05) among different treatments.

**Figure 3 antioxidants-12-00357-f003:**
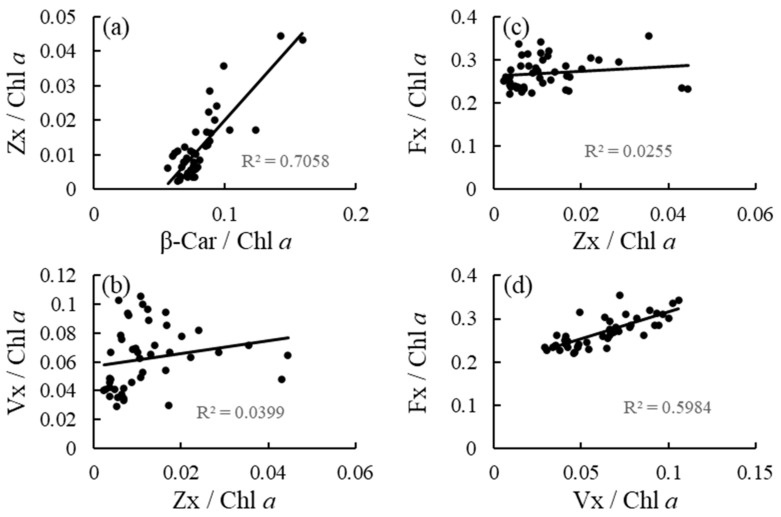
(**a**) Correlations between β-Car/Chl *a* and Zx/Chl *a* ratios, (**b**) between Zx/Chl *a* and Vx/Chl *a* ratios, (**c**) between Zx/Chl *a* and Fx/Chl *a* ratios, and (**d**) between Vx/Chl *a* and Fx/Chl *a* ratios of the brown alga *Sargassum fusiforme* acclimated to three irradiance levels (30, 130, and 300 μmol photons m^−2^ s^−1^) and three nutrient levels (0, 5, and 25% Provasoli’s enriched seawater, PESI).

**Figure 4 antioxidants-12-00357-f004:**
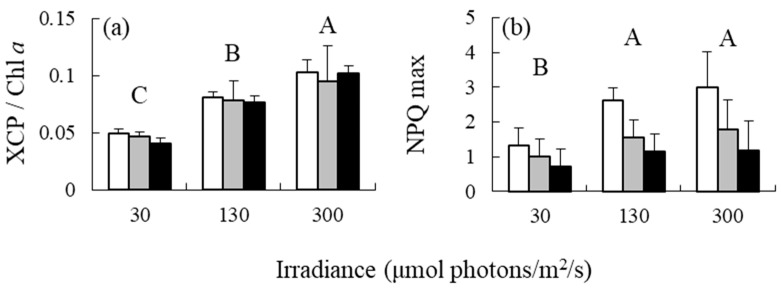
(**a**) Xanthophyll cycle pigment (XCP)/Chl *a* ratios and (**b**) non-photochemical quenching (NPQ) values of the brown alga *Sargassum fusiforme* acclimated to three irradiance levels (30, 130, and 300 μmol photons m^−2^ s^−1^) and three nutrient levels (0, 5, and 25% Provasoli’s enriched seawater, PESI). Values are expressed as means ± standard deviation. Different large letters indicate statistically significant differences (*p* < 0.05) among different treatments.

**Table 1 antioxidants-12-00357-t001:** Results of two-way ANOVA on the effects of irradiance and nutrients on the variables of *Sargassum fusiforme* holdfasts. MS, *F* and *p* indicate mean square, *F* value, and significance probability, respectively. Asterisks indicate the significant effects (*p* < 0.05).

	Irradiance			Nutrients			Irradiance * Nutrients		
Variables	MS	*F*	*p*	MS	*F*	*p*	MS	*F*	*p*
RGR	5.083	12.697	<0.001 *	1.282	3.201	0.053	0.273	0.681	0.609
*Fv*/*Fm*	0.122	13.886	<0.001 *	0.012	1.369	0.267	0.016	1.760	0.158
Chl *a*	0.199	56.553	<0.001 *	0.008	2.397	0.105	0.001	0.319	0.863
Chl *c*/Chl *a*	<0.001	0.895	0.417	<0.001	0.510	0.605	<0.001	1.028	0.406
β-Car/Chl *a*	0.010	1.301	0.285	0.003	0.420	0.660	0.011	1.404	0.252
Zx/Chl *a*	0.989	18.868	<0.001 *	0.016	0.307	0.737	0.127	2.414	0.067
Vx/Chl *a*	0.367	40.958	<0.001 *	0.019	2.104	0.137	0.001	0.143	0.965
Fx/Chl *a*	0.034	19.115	<0.001 *	<0.001	0.001	0.999	0.001	0.458	0.766
XCP/Chl *a*	0.449	73.771	<0.001 *	0.007	1.069	0.354	0.005	0.739	0.571
DES	0.014	1.769	0.185	0.010	1.199	0.313	0.017	2.162	0.093
NPQ	3.905	9.216	<0.001 *	6.439	15.196	<0.001	0.556	1.311	0.284

## Data Availability

The data and any statistical analysis are available from the corresponding author upon request.

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
