# Peer review of "Photoinhibition and Photoprotective Responses of a Brown Marine Macroalga Acclimated to Different Light and Nutrient Regimes"

_antioxidants, 2023, doi:10.3390/antiox12020357_

Round 1
Reviewer 1 Report
It is mentioned in the introduction that radiation stimulation will increase the antioxidant content in photosynthetic organisms, but in Figure 2 only β-carotene does not increase significantly, please explain the reason.
In line 218, the effect of irradiance on Fx/Chl a ratio was not detected in previous studies but irradiance exerted a positive effect on Fx/Chl a ratio in the present study. Please explain it.
In figure 1a, please explain why the RGR in 25% nutrient levels was lower than 5% nutrient levels at 30 irradiance levels.
In line 172, it is mentioned that irradiance had a significant effect on NPQ, and in figure4, it also referred to nutrients levels had an effect on NPQ, please explain the reason that no significant interaction was observed between irradiance and nutrient levels.
Author Response
Responses to the comments from Reviewer 1
Reviewer 1: It is mentioned in the introduction that radiation stimulation will increase the antioxidant content in photosynthetic organisms, but in Figure 2 only β-carotene does not increase significantly, please explain the reason.
Reviewer 1: In line 218, the effect of irradiance on Fx/Chl a ratio was not detected in previous studies but irradiance exerted a positive effect on Fx/Chl a ratio in the present study. Please explain it.
HE: Thank you for the comments. In order to discuss these reasons, the 5th paragraph of discussion was drastically revised as below.
Before revision: As one of the pigments involved in the carotenoid biosynthetic pathway that leads to the formation of xanthophylls from carotene, β-Car can be converted to β-cryptoxanthin, Zx, Ax, Vx, and neoxanthin, while Fx in brown algae is hypothesized to be synthesized from neoxanthin [8]. The hypothesis is partially supported by the re-sults of the present study, which showed a strong correlation between Vx/Chl a and Fx/Chl a ratios of S. fusiforme holdfasts acclimated to nine types of treatments. In addition, a strong correlation was observed between β-Car/Chl a and Zx/Chl a ratios, whereas no correlation was observed between Zx/Chl a and Vx/Chl a ratios, implying that the conversion rates of β-Car to Zx and Vx to Fx were higher than the epoxidation rate of Zx to Vx.
After revision: In the carotenoid biosynthetic pathway, β-Car can be converted to β-cryptoxanthin, Zx, Ax, Vx, and neoxanthin, while Fx in brown algae is hypothesized to be synthesized from neoxanthin [6]. The hypothesis is partially supported by the results of the present study, which showed a stronger correlation between Vx/Chl a and Fx/Chl a ratio than between Zx/Chl a and Vx/Chl a or Fx/Chl a ratios in S. fusiforme holdfasts acclimated to nine types of treatments. In addition, a strong correlation was observed between β-Car/Chl a and Zx/Chl a ratios, nevertheless elevated irradiance did not affect β-Car/Chl a ratio but affected Zx/Chl a ratio, implying that β-Car might be rapidly converted to Zx in this alga. Xie et al. [12] reported that elevated irradiance increased β-Car content of the red alga Neopyropia yezoensis, while such an increase in β-Car/Chl a ratio was not observed in the brown alga S. fusiforme in the present study. In contrast, irradiance elevation did not affect the Fx/Chl a ratio of U. pinnatifida and S. fusiforme shoots but increased the ratio of S. fusiforme holdfasts in the present study. Hence, the chemical form of carotenoids accumulated as a result of high light acclima-tion may differ among species and parts within the species.
Reviewer 1: In figure 1a, please explain why the RGR in 25% nutrient levels was lower than 5% nutrient levels at 30 irradiance levels.
HE: Nutrient absorption is reported to be affected by irradiance and therefore the effect of nutrient enrichment on growth might be weakened by decreased irradiance in the present study. However, the nutrient effect on the growth was not significant in the present study. In this case, we think we should not discuss this further.
Reviewer 1: In line 172, it is mentioned that irradiance had a significant effect on NPQ, and in figure4, it also referred to nutrients levels had an effect on NPQ, please explain the reason that no significant interaction was observed between irradiance and nutrient levels.
HE: Significant interactions between irradiance and nutrient conditions mean that the effect of irradiance varies with changing nutrient conditions and vice versa. We think we should discuss the reason when such interactions were rarely detected, but we cannot discuss the reasons of no significant interactions, which is commonly found.
Reviewer 2 Report
The manuscript presents a straightforward study of light and nutrient responses in brow algae, in particular focusing on pigment ratios, photosynthetic factors, and growth. The study found that light intensity did influence many of these factors, but did not see much response to nutrient levels, despite this combinatorial approach being a major focus of the work. As such, the work does not appear to present any new information advancing the field beyond that presented in the cited articles.
Notable aspects of the presentation are not well-described or put into context with the results, which could impact how novel this work appears to the reader. The stated intention to focus on pigment ratios versus Chl a is valid (and common), but the authors did not adequately address why this approach was used and a hint that raw values were variable (line 118) requires further explanation. Similarly, the work does not explain why xanthophyll cycle pigment ratios were not assessed. Given that Zx to Vx cycles in response to light stress, and that the work found higher overall xanthophyll pigments in higher light, it would be informative to know whether these increases also saw a change in Zx to Vx ratios.
Overall, the work could use many substantive changes to the discussion and explanation of how this work advances knowledge in the field. As such, the results appear either incremental or repetitive from previous studies.
Author Response
Responses to the comments from Reviewer 2
Reviewer 2: The manuscript presents a straightforward study of light and nutrient responses in brow algae, in particular focusing on pigment ratios, photosynthetic factors, and growth. The study found that light intensity did influence many of these factors, but did not see much response to nutrient levels, despite this combinatorial approach being a major focus of the work. As such, the work does not appear to present any new information advancing the field beyond that presented in the cited articles.
Reviewer 2: Overall, the work could use many substantive changes to the discussion and explanation of how this work advances knowledge in the field. As such, the results appear either incremental or repetitive from previous studies.
HE: Thank you for the comments. In order to emphasize the novelty of the present study, introduction was drastically revised as below.
Before revision: In photosynthetic organisms, high irradiance (i.e., high light intensity) under environmental stresses, such as low and high temperature, often causes photoinhibition, which is described as a decline in photosystem II efficiency (Fv/Fm), because excess light energy enhances the production of harmful reactive oxygen species in chloroplasts [1-3]. To cope with the oxidative stress, such organisms have evolved several photoprotective mechanisms, including the production of antioxidants and dissipation of the excess light energy as heat [2-4]. The heat dissipation can be quantified as non-photochemical quenching (NPQ) using pulse amplitude modulation (PAM) chlorophyll fluorometer [4-6]. Both mechanisms are strongly associated with carotenoids (carotenes and xanthophylls) because several compounds, such as β-carotene (β-Car), zeaxanthin (Zx), and fucoxanthin (Fx) act as antioxidants [3,7,8], and the de-epoxidation of xanthophyll cycle pigments (XCP) from violaxanthin (Vx) to Zx through antheraxanthin (Ax) is one of major drivers of NPQ in plants and brown algae [4-6]. Carotenoid-to-chlorophyll (Chl) a ratio and NPQ of various organisms increase in response to high light acclimation [9-13]. Moreover, NPQ increases under nutri-ent-deficient conditions in plants [14,15] and algae [16-18], although the changes in XCP have rarely been quantified [14]. The photoprotective responses of plants and al-gae are thought to be enhanced by elevated irradiance combined with reduced nutrient availability.
Large brown algae (kelps and fucoids) are the dominant taxa in temperate reef ecosystems [19-21]. Brown algae contain Chl a, Chl c, XCP, and Fx, as well as the other heterokont members [4,22,23]. Fx binds with Chl a and c [23,24], and acts as an anti-oxidant [8]. However, the role of Fx as a photoprotective compound in brown algae remains unclear because the Fx/Chl a ratio does not change under elevated irradiance acclimation [25,26]. In addition, NPQ in brown algae is reported to depend on the total amount of XCP (i.e., XCP pool size) and de-epoxidation states (DES) of XCP, which can be calculated as a ratio of de-epoxidated XCP during light exposure (0.5Ax+Zx) to total XCP [4,10]. Previous studies have shown the effects of elevated irradiance and nutrient availability on brown algal pigments including Chl a content, Chl c/ Chl a, Fx/Chl a, and XCP/Chl a ratios [25,26]. Moreover, several studies reported that NPQ of brown algae increased under high light acclimation and reduced nutrient availability [27-28]. However, a few studies have studied the combined effects of acclimation irradiance and nutrient condition on both NPQ and XCP/Chl a.
After revision: As one of the pigments involved in the carotenoid biosynthetic pathway that leads to the formation of xanthophylls from carotene, β-Car can be converted to β-cryptoxanthin, Zx, Ax, Vx, and neoxanthin, while Fx in brown algae is hypothesized to be synthesized from neoxanthin [6]. Some of these carotenoids bind with protein and chlorophyll (Chl) a, constituting photosystem II reaction center [9] and light-harvesting complex II [10]. The carotenoid-to-Chl a ratio of various organisms is reported to increase in response to high light acclimation [11-13]. Moreover, our pre-vious studies have shown the effects of elevated irradiance and nutrient availability on Chl a content, Chl c/ Chl a, Fx/Chl a, and XCP/Chl a ratios in brown algae [14,15]. However, the role of Fx as a photoprotective compound in brown algae remains un-clear because the Fx/Chl a ratio does not change under elevated irradiance acclimation [14,15].
The heat dissipation can be quantified as non-photochemical quenching (NPQ) using pulse amplitude modulation (PAM) chlorophyll fluorometers [4–6]. Previous studies have shown that high light acclimation enhanced NPQ (or the similar indicator, qN) of several organisms, including brown algae [4,16–18]. Moreover, NPQ is reported to increase under nutrient-deficient conditions in plants [19–20] and algae [17,21–23], although the underlying physiological mechanisms remain uncertain [14]. Meanwhile, NPQ generally correlates with de-epoxidation states (DES) of XCP in plants [24] and brown algae [11], and it is also strongly affected by the total amount of XCP (i.e., XCP pool size) in brown algae [4, 11]. Hence, increased NPQ of brown algae under reduced nutrient availability may be caused by changing XCP pool size and/or DES of XCP. However, the nutrient effects on NPQ and XCP of brown algae have rarely been quan-tified at the same time.
Reviewer 2: Notable aspects of the presentation are not well-described or put into context with the results, which could impact how novel this work appears to the reader. The stated intention to focus on pigment ratios versus Chl a is valid (and common), but the authors did not adequately address why this approach was used and a hint that raw values were variable (line 118) requires further explanation. Similarly, the work does not explain why xanthophyll cycle pigment ratios were not assessed. Given that Zx to Vx cycles in response to light stress, and that the work found higher overall xanthophyll pigments in higher light, it would be informative to know whether these increases also saw a change in Zx to Vx ratios.
HE: Thank you for the comments. In order to explain why we used carotenoid-to-Chl a ratios, we added an explanation that “some of these carotenoids bind with protein and chlorophyll (Chl) a, constituting pho-tosystem II reaction center [9] and light-harvesting complex II [10]” in the 2nd paragraph of “Introduction”. Regarding de-epoxidation state (DES) of XCP, which can be calculated as the ratio of 0.5Ax+Zx to total XCP, it was not affected by irradiance and nutrient conditions in the present study (Table 1). We added the explanation how to caluculate it in “Materials and methods”.
Reviewer 3 Report
line
50-51 «Brown algae contain Chl a, Chl c, XCP, and Fx, as well as the other heterokont members [4,22,23].» The sentence should be reformulated.
91 The kind of light source should be mentioned. What is the spectral range for the μmol photons m-2 s-1 values?
132 Explain what MS, F, and P mean. «df» should not need to be explained,
Consider citing the following paper:
Xu, L., Luo, L., Zuo, X. et al. (2022) Effects of temperature and irradiance on the regeneration of juveniles from the holdfasts of Sargassum fusiforme, a commercial seaweed. Aquaculture 557: 738317.
Some formalities:
The authors sometimes write «μmol photons m-2 s-1)», sometimes «photon» instead of «photons».
Scientific species names are sometimes italicized, sometimes not.
I suppose that journal names should be abbreviated in the reference list.
«a» in «Chl a» should probably be in italics; the coresponding for «Chl d».
Author Response
Responses to the comments from Reviewer 3
Reviewer 3: Line 50-51 «Brown algae contain Chl a, Chl c, XCP, and Fx, as well as the other heterokont members [4,22,23].» The sentence should be reformulated.
HE: Thank you for the comments. This sentence was deleted in the process of modification.
Reviewer 3: 91 The kind of light source should be mentioned. What is the spectral range for the μmol photons m-2 s-1 values?
HE: We added an explanation that “light with wavelength of 400–700 nm (i.e., photosynthetically active radiation) was provided by white fluorescent tubes” in the 3rd paragraph of “Materials and methods” We confirmed the wavelength of the light, although we cannot show the data in this manuscript because we are going to use it in the other article.
Reviewer 3: 132 Explain what MS, F, and P mean. «df» should not need to be explained.
HE: We added the explanation in the legend of Table 1.
Reviewer 3: Consider citing the following paper:
Xu, L., Luo, L., Zuo, X. et al. (2022) Effects of temperature and irradiance on the regeneration of juveniles from the holdfasts of Sargassum fusiforme, a commercial seaweed. Aquaculture 557: 738317.
HE: We cited this article as below (reference number is [29]).
The 4th paragraph of “Introduction”: Large brown algae (kelps and fucoids) are the dominant taxa in temperate reef ecosystems [25–27]. The present study investigated the combined effects of irradiance and nutrient availability on photosynthetic parameters (Fv/Fm and NPQ) and pig-ments (Chl a content, Chl c/ Chl a ratio, β-Car/Chl a ratio, Fx/Chl a ratio, XCP/Chl a ra-tio, and DES) in the fucoid brown alga Sargassum fusiforme, which is one of the com-mon and commercially important species in Asian countries. Holdfasts of the species were used in the present study instead of the shoots because they have important eco-logical traits, including the ability to grow without shoots, regeneration ability via germination of new shoots (i.e., vegetative reproduction) [28, 29], and high tolerance to warm and nutrient-poor conditions [28].
The 2nd paragraph of “Discussion”: The maximum photosystem II efficiency (Fv/Fm) of macroalgae generally decreases diurnally under strong irradiance around midday and recovers during the afternoon (i.e., dy-namic photoinhibition, [33]). Bischof et al. [34] reported that high light acclimation reduced the degree of photoinhibition and increases the rate of recovery, although long-term exposure to stronger irradiance may cause a gradual decrease in Fv/Fm (i.e. chronic photoinhibition, [33]). In fact, Endo et al. [36] reported that Fv/Fm of the kelp Eisenia bicyclis decreased under elevated irradiance (30–180 μmol photons m-2 s-1) but remained unchanged under nutrient enrichment (using 25% PESI) conditions. Xu et al. [29] also showed that Fv/Fm of S. fusiforme reduced in response to increased irradiance (70–140 μmol photons m-2 s-1) at a high temperature of 24 ℃ but not at 16–20 ℃. A similar observation was made for S. fusiforme in the present study, suggesting that high light acclimation at 130–300 μmol photons m-2 s-1 often causes photoinhibition in these brown algal species.
Reviewer 3: The authors sometimes write «μmol photons m-2 s-1)», sometimes «photon» instead of «photons».
HE: We revised them.
Reviewer 3: Scientific species names are sometimes italicized, sometimes not.
HE: We revised them.
Reviewer 3: I suppose that journal names should be abbreviated in the reference list.
HE: We revised them.
Reviewer 3: «a» in «Chl a» should probably be in italics; the coresponding for «Chl d».
HE: We revised them.
Round 2
Reviewer 1 Report
The author has made improvements in response to the review comments, and I have no further questions
Author Response
Reviewer 1: The author has made improvements in response to the review comments, and I have no further questions.
HE: Thank you very much.

Reviewer 2 Report
No new comments. Previous major concerns not addressed.
Author Response
Reviewer 2: No new comments. Previous major concerns not addressed.
(Round 1): Overall, the work could use many substantive changes to the discussion and explanation of how this work advances knowledge in the field. As such, the results appear either incremental or repetitive from previous studies.
HE: We revised the discussion drastically as below. We will ask an English editing company to correct the grammar after acceptance.
Discussion
The indicator of photoinhibition, Fv/Fm of photosynthetic organisms is known to decrease in response to high irradiance under environmental stresses, such as high or low temperature [1-3]. For example, Balfagón et al. [36] showed that a negative effect of increased irradiance (50–600 μmol photons m-2 s-1) on Fv/Fm of the terrestrial plant Arabidopsis thaliana was synergized by elevated temperature from 23 ℃ to 42 ℃. Simi-larly, Xu et al. [29] reported that Fv/Fm of the brown algae S. fusiforme reduced in re-sponse to increased irradiance (70–140 μmol photons m-2 s-1) at a high temperature of 24 ℃ but not at 16–20 ℃. Machalek et al. [37] also reported that Fv/Fm of the kelp Sac-charina latissima decreased in response to elevated irradiance (15–150 μmol photons m-2 s-1) at a low temperature of 5 ℃ but not at 17 ℃. Meanwhile, nutrient-poor condi-tions during summer is one of environmental stresses for brown algae, including S. fu-siforme holdfasts [28, 38]. In the present study, Fv/Fm of S. fusiforme holdfasts decreased by elevated irradiance (30–300 μmol photons m-2 s-1), while no significant interaction between irradiance and nutrient conditions was observed. This result suggests that the negative effect of high irradiance on Fv/Fm of this species was not strengthened by nu-trient-poor conditions.
Balfagón et al. [36] also showed that the increased irradiance combined with the elevated temperature caused not only decreased Fv/Fm but also leaf senescence and reduced survival rate of A. thaliana. Similarly, Endo et al. [39] reported that elevated temperature (23–26 ℃) at high irradiance (180 μmol photons m-2 s-1) resulted in re-duced Fv/Fm and blade erosion of the kelp Eisenia bicyclis. Thus, photoinhibition is of-ten accompanied by physiological damages in plant and algal bodies. However, the elevated irradiance (30–300 μmol photons m-2 s-1) reduced Fv/Fm but enhanced growth of S. fusiforme holdfasts in the present study. Therefore, the decline of Fv/Fm under high light acclimation seem not necessarily accompany growth suppression in the absence of environmental stresses.
Chl a content of plants and algae is known to decrease under high light acclima-tion [14,15,40,41] and this response contributes to limiting absorption of excess light energy [19]. Also in the present study, Chl a content of S. fusiforme holdfasts decreased in response to increased acclimation irradiance from 30 to 300 μmol photons m-2 s-1. Moreover, microalgae are known to decrease their ratio of accessory Chl to Chl a under high light acclimation [41]. Previous studies have shown that Chl c/Chl a ratio of brown algae was lower in the thalli grown at shallower depths or sun-exposed sites than the same species collected at greater depths or shaded sites [42]. However, in the present study, no significant difference was found in Chl c2/Chl a ratio of S. fusi-forme holdfasts acclimated in three different irradiance conditions. Charan et al. [15] also reported that Chl c2/Chl a ratio of S. fusiforme shoots was not affected by irradiance at 23 ℃, although the value decreased in response to increased irradiance (30–150 μmol photons m-2 s-1) combined with heat stress (26 ℃). Moreover, Endo et al. [14] re-ported that increased irradiance (30–180 μmol photons m-2 s-1) caused a decrease in Chl c1/Chl a ratio but an increase in Chl c2/Chl a ratio of the kelp U. pinnatifida. Hence, the variation of Chl c/Chl a ratio of brown algae along depth and sun-exposure gradi-ents found in previous studies [42] cannot be explained by the response to changing irradiance. It might be affected by changes in light quality associated with depth or sun-exposure, although this hypothesis needs to be tested.
In the carotenoid biosynthetic pathway, β-Car can be converted to β-cryptoxanthin, Zx, Ax, Vx, and neoxanthin, while Fx in brown algae is hypothesized to be synthesized from neoxanthin [6]. The hypothesis is partially supported by the re-sults of the present study, which showed a stronger correlation between Vx/Chl a and Fx/Chl a ratio than between Zx/Chl a and Vx/Chl a or Fx/Chl a ratios in S. fusiforme holdfasts acclimated to nine types of treatments. In addition, a strong correlation was observed between β-Car/Chl a and Zx/Chl a ratios, nevertheless elevated irradiance did not affect β-Car/Chl a ratio but affected Zx/Chl a ratio, implying that β-Car might be rapidly converted to Zx in this alga. Xie et al. [12] reported that elevated irradiance increased β-Car content of the red alga Neopyropia yezoensis, while such an increase in β-Car/Chl a ratio was not observed in the brown alga S. fusiforme in the present study. In contrast, irradiance elevation did not affect the Fx/Chl a ratio of U. pinnatifida and S. fusiforme shoots but increased the ratio of S. fusiforme holdfasts in the present study, suggesting that Fx is one of photoprotective compounds in this alga. Hence, the chemical form of carotenoids accumulated as a result of high light acclima-tion may differ among species and parts within the species.
Ocampo-Alvarez et al. [11] observed that the blades of the kelp Macrocystis pyr-ifera acclimated to shallow depths exhibited higher XCP/Chl a ratios and NPQ than those grown at greater depths. Moreover, previous studies have reported that high light acclimation resulted in increases in XCP/ Chl a ratios [14] and NPQ [18] of brown algae. Also in the present study, the elevated irradiance increased both XCP/Chl a ratios and NPQ of S. fusiforme holdfasts. Hence, increased NPQ under high light acclimation can be explained by increased XCP pool size of the brown algae. Meanwhile, NPQ decreased in response to nutrient enrichment in S. fusiforme, which is consistent with the observation made in plants [19,20] and algae [17, 21–23]. Although XCP/Chl a ratio can decrease under nutrient enriched conditions [14, 15], XCP/Chl a ratios and DES observed in our study did not change. The results suggest that the decrease in NPQ in response to nutrient enrichment was not due to the decreases in XCP pool size and de-epoxidation rate of Vx to Zx. Therefore, NPQ can be used as a sensitive stress marker for nutrient deficiency but cannot be used to estimate XCP pool size and state. Such a reduction in NPQ without any changes in XCP might be associated with state transitions between photosystems I and II [43,44], although this possibility needs to be examined.
- Conclusions
The results of the present study showed that elevated irradiance caused photoin-hibition and Chl a degradation but enhanced growth and photoprotection, which were reflected by the increases in Fx/Chl a ratios, XCP/Chl a ratios, and NPQ in S. fusiforme holdfasts, while nutrient availability had negligible effects on the variables other than NPQ. These traits may enhance the survival and growth of S. fusiforme holdfasts, without forming a canopy of its own shoots under high irradiance and nutrient-poor conditions that occur during summer [28]. However, little is known regarding the combined effects of elevated summer temperatures under climate change and in-creased irradiance on XCP/Chl a ratios and NPQ in brown algae, including S. fusiforme. Furthermore, the mechanisms of associated with decreases in NPQ under nutrient en-richment conditions still remain unknown. Therefore, further studies should be con-ducted to investigate the effects of abiotic factors on NPQ and associated variables to enhance our understanding of photoprotection in photosynthetic organisms, including brown macroalgae.